# Ablation of Ghrelin Receptor Mitigates the Metabolic Decline of Aging Skeletal Muscle

**DOI:** 10.3390/genes13081368

**Published:** 2022-07-30

**Authors:** Colleen O’Reilly, Ligen Lin, Hongying Wang, James Fluckey, Yuxiang Sun

**Affiliations:** 1Department of Health and Kinesiology, Texas A & M University, College Station, TX 77843, USA; colleen.l.oreilly@tamu.edu; 2USDA/ARS Children’s Nutrition Research Center, Department of Pediatrics, Baylor College of Medicine, Houston, TX 77030, USA; ligenl@um.edu.mo; 3State Key Laboratory of Quality Research in Chinese Medicine, Institute of Chinese Medical Sciences, University of Macau, Macau 999078, China; 4Department of Nutrition, Texas A & M University, College Station, TX 77843, USA; hongying.wang@ag.tamu.edu

**Keywords:** GHS-R, aging, irisin, skeletal muscle

## Abstract

The orexigenic hormone ghrelin has multifaceted roles in health and disease. We have reported that ablation of the ghrelin receptor, growth hormone secretagogue receptor (GHS-R), protects against metabolic dysfunction of adipose tissues in aging. Our further observation interestingly revealed that GHS-R deficiency phenocopies the effects of myokine irisin. In this study, we aim to determine whether GHS-R affects the metabolic functions of aging skeletal muscle and whether GHS-R regulates the muscular functions via irisin. We first studied the expression of metabolic signature genes in gastrocnemius muscle of young, middle-aged and old mice. Then, old GHS-R knockout (*Ghsr^−/−^*) mice and their wild type counterparts were used to assess the impact of GHS-R ablation on the metabolic characteristics of gastrocnemius and soleus muscle. There was an increase of GHS-R expression in skeletal muscle during aging, inversely correlated with the decline of metabolic functions. Remarkedly the muscle of old GHS-R knockout (*Ghsr^−/−^*) mice exhibited a youthful metabolic profile and better maintenance of oxidative type 2 muscle fibers. Furthermore, old *Ghsr^−/−^* mice showed improved treadmill performance, supporting better functionality. Also intriguing to note was the fact that old GHS-R-ablated mice showed increased expression of the irisin precursor *FNDC5* in the muscle and elevated plasma irisin levels in circulation, which supports a potential interrelationship between GHS-R and irisin. Overall, our work suggests that GHS-R has deleterious effects on the metabolism of aging muscle, which may be at least partially mediated by myokine irisin.

## 1. Introduction

Aging is associated with a procreative reduction in lean body mass, and consequent losses of physical strength and mobility, leading to decreased quality of life. Sarcopenia, the loss of muscle mass during aging, has been directly linked to increased mortality [1]. While skeletal muscle has a primary role in locomotion and maintenance of posture, it is also critical in organism-wide metabolic homeostasis. The loss of strength and function of skeletal muscle with advancing age often further exacerbates the metabolic dysfunctions, including mitochondrial dysfunction [2,3,4], glucose intolerance and insulin resistance [4,5,6], and anabolic resistance [7,8,9]. Another feature of advanced aging is a shift in skeletal muscle fiber type from faster to slower phenotypes [10,11,12]. Although not completely understood, many studies suggest metabolic dysregulation of muscle has a major role in leading to the deleterious progression of sarcopenia in aging.

Ghrelin is an acylated 28 amino acid peptide predominantly produced in the X/A-like enteroendocrine cells of the stomach [13]. Ghrelin, mainly known for its orexigenic effects, is now increasingly recognized as a key regulator of nutrient sensing, energy and glucose homeostasis, and aging metabolism. Currently, the only known biologically relevant receptor of ghrelin is growth hormone secretagogue receptor-1a (GHS-R1a), and its canonical function is to exert the orexigenic effect via the hypothalamus [13,14]. GHS-R1a is known to be constitutively active [15,16,17] and has been suggested to also have noncanonical functions [18,19]. Previous research from our lab and others have shown that knockout of the GHS-R gene in mice protects against diet-induced obesity [20], aging insulin resistance [21], as well as obesity and aging-associated inflammation [22,23].

In addition to its well-known orexigenic property, our work suggests that ghrelin signaling through GHS-R has an important role in adipose tissue metabolism, specifically, GHS-R ablation protects against thermogenic impairment in aging [21,24] and that GHS-R works as a metabolic thermostat in brown adipose tissue [25]. Enhancing thermogenesis is considered a promising new strategy to prevent obesity and metabolic syndrome. Irisin is a myokine that is known to activate browning or beiging of adipose tissue; thus, irisin is also called a thermogenic adipo-myokine [26] due to its significant metabolic implications. Irisin is produced by the cleavage of fibronectin type III domain-containing protein 5 (*FNDC5*) gene, it is peroxisome proliferator-activated receptor-γ coactivator-1α (PGC-1α)-dependent [27]. It is known that skeletal muscle is a major source of irisin in the circulation, and its level is dramatically elevated by exercise [28]. Intriguingly, in vitro skeletal muscle cells treated with irisin exhibited increased glucose uptake [29] and increased oxidative metabolism [29,30,31], and the effects were suggested to be mediated by master metabolic regulator AMPK [29,31]. Despite the promising in vitro evidence, irisin’s functions in muscle in vivo are largely unclear, and the understanding of its function in aging muscle is even more scarce. The purpose of this work was to determine the role of GHS-R in aging muscle, and to assess whether irisin mediates the effect of GHS-R in muscle metabolism in aging.

## 2. Materials and Methods

### 2.1. Animals

*Ghsr^−/−^* mice in C57BL/6J background were generated as we previously described [21,32]. Animals were housed under controlled temperature and lighting (75 ± 1 F; 12 h light/dark cycle) with free access to food and water. All animals were fed the same rodent chow diet. Data-relevant age cohorts were developed as previously described [21], and the age groups were described as young (4–5 m), middle-aged (12–14 m) and old (18–26 m). Animals of 6–9 were used in various experiments. All experiments were approved by the Animal Care Research Committee at the Baylor College of Medicine.

### 2.2. Real-Time RT-PCR

Real-time RT-PCR was performed on both gastrocnemius and soleus muscles as previously described [21,22]. Briefly, total RNA was isolated using TRIzol reagent (Invitrogen, Carlsbad, CA, USA) following the manufacturer’s instructions. RNA was treated with DNAse and then run on gels to validate purity and quality. The cDNA was synthesized from 1 µg RNA using the Superscript III First-Strand synthesis system for RT-PCR (Invitrogen). Real-time RT-PCR was performed on ABI 7900 using the SYBR green PCR master mix or the TaqMan gene expression master mix (Invitrogen). *18S* RNA and *β-actin* were used as internal controls. All primer and probe information are available upon request.

### 2.3. Lipid Content

Lipid content of the gastrocnemius muscle was measured as previously described [33]. Gastrocnemius muscle samples (about 50 mg) were minced in liquid nitrogen and transferred to ice-cold Teflon-lined screw-cap tubes. Then, 1 mL of chloroform: methanol (2∶1 *v/v*) mixture was added to each tube. The samples were than homogenized in a sonicator for 5 s, and the tubes were placed in a rotary mixer at room temperature for 24 h. The lower organic phase was transferred to another tube and washed with PBS twice. After evaporation, precipitate was weighed. The lipid content was normalized by tissue weight.

### 2.4. Western Blotting

Western blot analysis was performed on gastrocnemius muscle as described [34,35]. Briefly, whole tissue was pulverized with liquid nitrogen and then 40 mg of tissue was homogenized in cold Norris buffer [1:10 tissue/buffer (mg/μL); 25 mM HEPES, 5 mM β-glycerophosphate, 200 μM ATP, 25 mM benzamidine, 2 mM PMSF, 4 mM EDTA, 10 mM MgCl2, 100 mM NF, 10 mM Na3VO4, Sigma protease inhibitor cocktail P8340 (Sigma-Aldrich, St. Louis, MO, USA), and 1% TritonX100, pH 7.4]. Homogenates were then spun at 14,000 RPM for 30 min at 4 °C to separate myofibrillar rich fractions from cytosolic rich fractions. The cytosolic fractions were denatured in 4x laemmli buffer at 95 °C and identical quantities of protein were loaded onto 8% polyacrylamide gels. Following 1.5 h electrophoresis at 20 mA in standard buffer, a semi-dry 1 h transfer (7.5 mA/cm^2^) was used to transfer proteins onto 0.2 μM PVDF membranes soaked in methanol.

Membranes were then blocked for 1 h in blocking buffer (5% dried milk (*w/v*) in Tris Buffered Saline) and incubated overnight in a heat-sealed plastic bag containing 1:1000 primary antibody/buffer (5% BSA (*w/v*) in TBS). Blots were probed with phospho-AMPK (Cell Signaling, #2531), total AMPK (Cell Signaling, #5832), phospho ACC (Cell Signaling, #11818), total ACC (Cell Signaling, #3662), UCP3 (Cell Signaling, #97000) and Glut 4 (Cell Signaling, #2213) antibodies. After a serial wash step 1xTBS (3 × 5 min), membranes were incubated for 1 h at room temperature with 1:2000 secondary antibody/buffer (5% milk (*w/v*) in TBS. After another serial wash step membranes were incubated for 5 min in ECL (Pierce) and bands were developed with a CCD camera mounted in a FluorChem SP imaging system (Alpha Innotech, San Leandro, CA, USA). Optical Density was determined using the Studio Lite (LI-COR Biosciences, Lincoln, NE, USA) and was automatically set to subtract nonspecific binding from densitometry values. All bands were normalized to total protein from Panceau S staining and expressed as arbitrary units. 

### 2.5. Fiber Type Analysis

Fiber type analysis of gastrocnemius muscle was completed using SDS-PAGE and silver staining as previously described with modifications [36,37,38]. Briefly, the myofibrillar rich pellets obtained from the 40 mg of tissue were resuspended in 300 μL of Norris Buffer and homogenized. An aliquot of the resuspended myofibrillar fraction was denatured with 4× laemmli buffer at 95 °C and 2 μL was applied to 8% polyacrylamide gels for 20 h at 40 V. Silver stain was completed using Pierce^TM^ Silver Stain Kit (Thermo Scientific 24612, Waltham, MA, USA) following the manufacturer’s instructions. Gels were imaged using Alpha Innotech imager (Alpha Innotech, San Leandro, CA, USA) and myosin heavy chains were identified according to their molecular weights as described previously [39]. The percentage of each myosin isoform was determined through densitometry with Image Studio^TM^ Lite software (LI-COR Biosciences, Lincoln, NE, USA).

### 2.6. Treadmill Endurance Test

A treadmill endurance protocol was performed using an Exer-3/6 open treadmill (Columbus Instruments, Columbus, OH, USA) similar to previously described [40]. Mice started the test at 6 m/min. Treadmill speed was then increased by 2 m/min every 2 min, until the mice were exhausted. Exhaustion was defined as spending more than 10 s on the shocker without attempting to re-enter the treadmill.

### 2.7. Plasma Irisin Content

To determine irisin content in plasma, a commercial IRSIN ELISA kit was used (EK-067-16, Phoenix Pharmaceuticals Inc., Burlingame, CA, USA). Samples were prepared according to manufacturer’s instructions, mouse plasma with a 5× dilution was used.

### 2.8. Statistical Analyses

One-way ANOVA was used to evaluate the significance of interaction between genotypes, and post hoc tests were used to follow up. When appropriate, two-tailed Student’s *t*-test was used to determine the statistical significance between genotypes. The results are expressed as mean ± standard error of the mean. Statistical significance was set as *p* < 0.05.

## 3. Results

### 3.1. Muscular Aging Is Postitively Correlated with Increased GHS-R Expression in Muscle

It has been previously shown that there is an increase of GHS-R in adipose tissue of aging mice, accompanied with metabolic dysfunctions [21,22,41]. To assess whether GHS-R expression is indeed correlated with metabolic disfunctions of skeletal muscle, GHS-R and common marker genes of mitochondrial function and glucose uptake were assessed in the skeletal muscle of young, middle-aged and old mice using real time RT-PCR (Figure 1). Relative expression of mitochondrial genes of uncoupling protein 3 (UCP3) and Sirtuin-1 (SIRT1) was significantly lower in the middle-aged and old mice, consistent with the known mitochondrial functional decline in aging muscle (Figure 1a). Additionally, the relative expression of PGC-1α, a potent stimulator of mitochondrial biogenesis and central mediator of energy metabolism, was also reduced in the aged mice (Figure 1a). Furthermore, insulin receptor substrate 1 (IRS1) and glucose transporter-4 (GLUT4) were also lower in the middle-age and old age groups when compared to young group, supporting the concept of metabolic functional decline in aging skeletal muscle (Figure 1b). Interestingly, there was also increased GHS-R expression during aging, and GHS-R expression in skeletal muscle of old mice was significantly higher than that of young mice (Figure 1c). These results indicate that there is an increase of GHS-R expression in aging muscle, and GHS-R expression is correlated with metabolic dysfunction of aging skeletal muscle.

### 3.2. Muscle of Old GHS-R Knockout Mice Reveals Improved Lipid Metabolism, Mitochondrial Function, and Insulin Sensitivity

Our previous work in ghrelin receptor knockout mice showed that GHS-R ablation reduces obesity and improves whole body insulin sensitivity in aging [21]. Here, we found that global ablation of GHS-R attenuates the decrements of mitochondrial and glucose uptake genes in muscles of the old mice (Figure 2). The gastrocnemius of the old *Ghsr^−/−^* groups had higher expression of UCP3 and PGC-1α, as well as a trend of increased expression of acetyl-CoA carboxylase 1 (ACC1). These mitochondrial genes are required for metabolic function and mitochondrial biogenesis. Consistently, lipid content (Figure 2b) of the gastrocnemius was reduced in the old *Ghsr^−/−^* mice when compared to old wild-type (WT) mice, suggesting an increase in β-oxidation in the muscle of *Ghsr^−/−^* mice. Additionally, the GLUT4 and IRS1 expression was also increased in the aged knockout mice (Figure 2c), supporting improved glucose uptake and insulin sensitivity in muscle of old *Ghsr^−/−^* mice.

We further analyzed the protein of some of the genes above, and trends of increased p-AMPK, p-ACC and UCP3 were detected in muscle of *Ghsr^−/−^* mice by Western blot analyses (Figure 3). The ratios of phosphorylated to total AMPK and ACC, as well as UCP3 content, all demonstrated higher trends, which is consistent with the elevated mRNA expression observed. Interestingly, Glut 4 total protein content was not different between the groups.

### 3.3. GHS-R Ablation Alters Expression of Myosin Heavy Chain of Skeletal Muscle, and Improves Treadill Performance of Old Mice

In the aging population, there is a consistent shift in skeletal muscle fiber type to a more oxidative fiber type, Type 2 to Type 1 [10,11,12]. In both gastrocnemius (Figure 4a) and soleus muscle (Figure 4b), mRNA content for MHC-IIa was higher in the old *Ghsr^−/−^* compared to WT. Fiber typing of the gastrocnemius myofibrillar rich fraction (Figure 4c) is consistent with our mRNA data, demonstrating that there is a phenotypic shift toward MHC-IIa, instead of MHC-IIb, in the muscle of aged *Ghsr^−/−^* mice (Figure 4c). Interestingly, in soleus muscle, MHC-I was lower with GHS-R ablation when compared to WT. Taken together, these results indicate a better maintenance of more oxidative (fast) fiber type in both gastrocnemius and soleus muscles of aged *Ghsr^−/−^* mice.

The observed mRNA alterations and fiber type analysis are further supported by the treadmill muscle functional test (Figure 4d). In the treadmill test, there increased trends of running time and distance traveled between the genotypes, and aged *Ghsr^−/−^* mice generated higher work output than that of the aged WT. The treadmill functional test is in agreement with the muscle fiber phenotype, which supports that GHS-R ablation improves muscle functionality.

### 3.4. Irisin/FNDC5 Expression Is Elevated in Skeletal Muslce of Old GHS-R Knockout Mice

Irisin, a cytokine found in multiple tissues, is the product of the gene *FNDC5*. Currently, skeletal muscle is touted as the highest expression site of *FNDC5*, and muscle is considered a major source of circulating irisin, thus irisin is considered a myokine. It is known that irisin can directly modulate skeletal muscle and other tissues, and irisin is associated with exercise performance due to its effects on mitochondrial function and glucose uptake [30,42]. Our results indicate that *FNDC5* is reduced in skeletal muscle of middle-aged and old mice compared to young animals (Figure 5a). Although we do not have *FNDC5* data in young and middle-aged *Ghsr^−/−^* mice, in old *Ghsr^−/−^* mice, we found that *FNDC5* mRNA expression in the gastrocnemius muscle increased and irisin levels in the circulation were elevated (Figure 5b,c). This result is in line with the improved metabolic profile observed in the muscle of aged *Ghsr^−/−^* mice.

## 4. Discussion

While it is clear that muscular aging often leads to losses of muscle mass and metabolic dysfunction, the understanding of how this progression occurs is still widely debated. Unlike ghrelin ubiquitously expressed, the expression of ghrelin receptor GHS-R is much more restricted in terms of both location and expression levels [41,43]. The restricted expression of GHS-R was previously reported [21,22,41]. Our current study demonstrated that GHS-R expression is increased in aging muscle, positively correlated with functional impairment, which suggests that ghrelin signaling may be involved in the functional decline of aging. Previous work in our lab has implicated that GHS-R is involved in the adiposity and insulin resistance that occurs with advanced age [21], but much of the our work was focused on adipose tissue [21,24,25]. The current work specifically focuses on the effects of GHS-R on muscular aging. We found that the ablation of GHS-R impacts the metabolic profile of skeletal muscle, showing an improved mitochondrial and glucose uptake expression profiles in aged skeletal muscle of GHS-R knockout mice.

A critical factor in aged skeletal muscle is the change in fiber composition to a relatively slow and more oxidative fiber type [10,11,12]. This appears to be from a preferential atrophy and eventual loss of type 2 muscle fiber types with aging [44,45]. In the present study, we found that the aged *Ghsr^−/−^* animals had higher MHC-IIa in both gastrocnemius and soleus muscle mRNA content, suggesting that the knockout of GHS-R mitigates the loss of these fiber types in aging. This is further supported by the fiber type analysis of the gastrocnemius, showing more oxidative MHC-IIa fiber type in the aged *Ghsr^−/−^* mice compared to that of WT mice. However, we also found that GHS-R deletion blocked the potential transition toward slower muscle types, evident in loss of type I mRNA in soleus muscle of *Ghsr^−/−^* mice. This suggests that the shift of fiber types and reduced loss of type 2 fibers with the ablation of GHS-R. Interestingly, a recent report showed similar results with ghrelin deletion, where aged ghrelin knockout mice exhibited an increased number of type 2a muscle fibers [46]. Our observation of muscle fiber type shift toward MHC-IIa was further supported by the treadmill function test, which showed the *Ghsr^−/−^* animals had a higher overall work output. It has been suggested that reprogramming of transcription factors is involved in mitochondrial biogenesis, linking it to adipose tissue browning/beiging [47], as well as a shift in fiber type [48,49,50]. Our data support that GHS-R ablation attenuates the aging-associated MHC-IIa fiber type shift.

This study also uncovered the relationship between GHS-R and irisin by studying skeletal muscle *FNDC5* and circulating irisin. First, we found that *FNDC5* is down-regulated in the skeletal muscle of old animals, suggesting that irisin may function as a pathogenic regulator for muscle aging. Findings from the current work are consistent with work by others showing lower serum irisin levels observed in middle-aged and old human subjects compared to their younger counterparts [28,51]. Second, we observed that the effect of GHS-R on skeletal muscle metabolism is correlated with irisin. Currently, irisin is considered a target molecule for obesity and insulin resistance, obesity and insulin resistance are common comorbidities of aging. While some recent studies have contended that irisin is linked to exercise and exercise impacts *FNDC5* [52,53,54], it is generally agreed that irisin is linked to the mass, strength and metabolism of skeletal muscle [52] and that irisin plasma circulation is the best known predictor of muscle mass in humans [28]. It has been reported that exogenous irisin has similar whole-body effect as that of free wheel running [53]. This concept is further supported by in vitro studies, where irisin increases gene expression for both glucose uptake and fatty acid oxidation [30,55], similar to our finding in GHS-R ablated old mice. This modulation has been proposed to be through AMPK phosphorylation and its downstream regulator such as PGC-1α [29,31]. Thus, irisin’s beneficial effects in skeletal muscle may be linked to GHS-R, and GHS-R acts as part of a feedback loop of GHS-R—irisin—AMPK—PGC-1α.

The best known effects of irisin are related to its impact on the browning or beiging of adipose tissues [28,56]. It was originally proposed that a proteosome cleaves irisin from *FNDC5* in the muscle, then irisin is released into the blood and subsequently reaches adipose tissues, where it activates thermogenesis [27,56]. Recently, however, the role of muscle as the main source of irisin has been called into question due to the observation that adipose tissue-expressing *FNDC5* appears to be more reflective of circulating irisin levels [52,54]. We have previously reported that *Ghsr^−/−^* mice have increased energy expenditure and enhanced thermogenesis [25] and that GHS-R knockdown in brown adipocytes activates thermogenic signaling [24], indicating that GHS-R is an important regulator of thermogenesis, and GHS-R has a cell-autonomous effect in brown adipose tissue. While the focus of the current study is not adipose tissue, our current work revealed that GHS-R ablation increases *FNDC5* in skeletal muscle and elevates circulating irisin in aged mice. Our data suggest that the increased *FNDC5* expression and elevated circulating irisin may activate AMPK signaling in the muscle of old *Ghsr^−/−^* mice. Indeed, it has been reported that that irisin alters glucose uptake in skeletal muscle cells via AMPK [29]. We observed that *FNDC5* expression in skeletal muscle is decreased in aging muscle opposite from that of GHS-R expression; moreover, *FNDC5* expression in skeletal muscle is increased in skeletal muscle of old *Ghsr^−/−^* mice and circulating irisin level is elevated in old *Ghsr^−/−^* mice. While the regulation of circulating irisin levels is most likely multifaceted, nonetheless, our new findings suggest that GHS-R is an important negative regulator of muscle metabolism in aging, and the effect of GHS-R in aging muscle may be mediated by irisin. While more research is warranted, the current data provide a potential new mechanism that connects nutrient sensing GHS-R to myosin irisin in muscle metabolism in aging.

While our novel observation that GHS-R regulates the irisin pathway in aging is very exciting, we recognize there are several limitations. Given that the current understanding of irisin is very fragmented and the concern that rodent and human models store and respond differently to irisin [28,57], more studies are needed to further elucidate the direct cause–effect relationship between GHS-R and irisin, and to determine whether the regulation of GHS-R on irisin is present in humans. While our present study does not provide a complete picture of how GHS-R regulates muscle metabolism in aging, our data do provide a novel insight for future inquiry into the crosstalk between nutrient sensing regulator and myosin irisin in muscular aging. We also want to note the following limitations of our current study: (1) We studied the expression of mitochondrial biogenesis and energy-metabolic genes in gastrocnemius muscle of young, middle-aged and old mice, but we were not able to conduct protein or activity assays to further solidify the result due to the tissue limitation of the old mice. (2) We studied a series of metabolic regulators in the skeletal muscle of old *Ghsr^−/−^* mice and their WT counterparts, and made the exciting observation that GHS-R has a pathogenic role in metabolic dysfunction of aging muscle. Unfortunately, we do not have sufficient tissue to further assess the consequences of these regulators in deciphering the hierarchy of the regulatory pathways. Future in-depth studies to validate GHS-R in muscle functionality throughout lifespan and aging muscle would significantly advance the understanding of ghrelin signaling in muscle aging and pathogenesis of sarcopenia. Future investigation of the cell-autonomous effect of GHS-R in myocytes by gene-knockdown or muscle-specific GHS-R knockout mouse model would be of great advantage. In addition, since irisin secretion is significantly increased by exercise, it would be very interesting to study the muscle metabolism of *Ghsr^−/−^* mice under exercise.

In conclusion, our study reveals for the first time that GHS-R is at least partially responsible for the metabolic decline of old skeletal muscle. The suppression of GHS-R mitigates the metabolic impairment of skeletal muscle in aging, at least partly mediated by irisin. Further studies are warranted to verify the exciting mechanistic network of GHS-R, irisin, and muscle metabolism.

## Figures and Tables

**Figure 1 genes-13-01368-f001:**
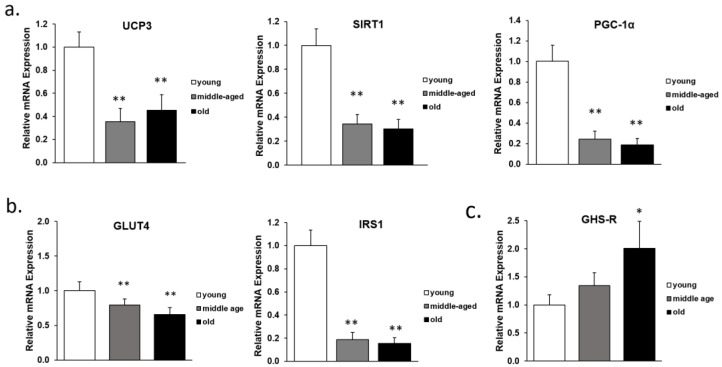
Relative mRNA expression in gastrocnemius muscle during aging. Young (4–5 m), middle-aged (12–14 m) and old (18–26 m) male C57BL/6 J mice were used. (**a**) Relative mRNA expression of mitochondrial functional markers of *UCP3*, *SIRT1* and *PGC-1α*; (**b**) relative gene expression of glucose transporter marker *GLUT* and insulin signaling marker *IRS1*; (**c**) relative gene expression of *GHS-R*. 18s and β-actin house-keeping genes were used as internal controls in qPCR analysis. One-way ANOVA analysis was used to compare middle-aged or old mice to young mice. Data are presented as means ± standard error. (*n* = 6). * *p* < 0.05, ** *p* < 0.001, middle-aged or old mice vs. young mice.

**Figure 2 genes-13-01368-f002:**
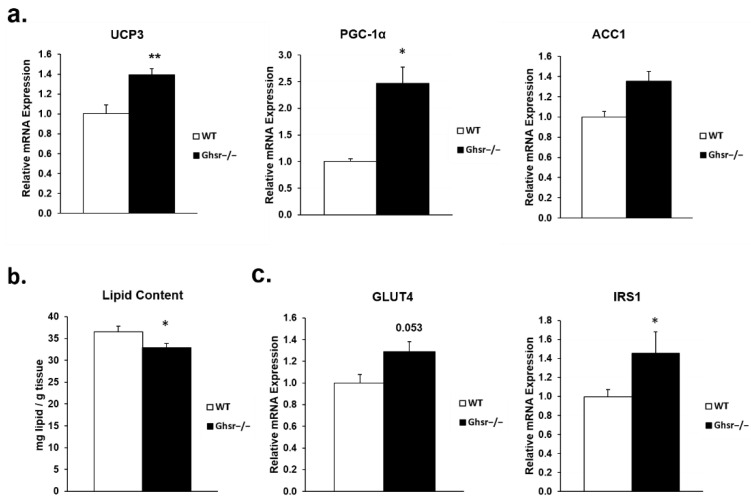
Effects of GHS-R ablation on metabolic dysfunction of aging muscle. Gastrocnemius muscle was from old (18–26 m) WT (open bar) and *Ghsr^−/−^* (black bar) mice. (**a**) *UCP3*, *PGC-1**α* and *ACC1* mRNA expression. (**b**) Lipid content in gastrocnemius muscle. (**c**) Relative expression of *IRS1* and *GLUT4* in gastrocnemius muscle. 18s and β-actin house-keeping genes were used as internal controls in qPCR analysis. Two-tailed Student’s *t*-tests were completed, and data are presented as means ± standard error (*n* = 9). * *p* < 0.05, ** *p* < 0.001, *Ghsr^−/−^.* vs. WT.

**Figure 3 genes-13-01368-f003:**
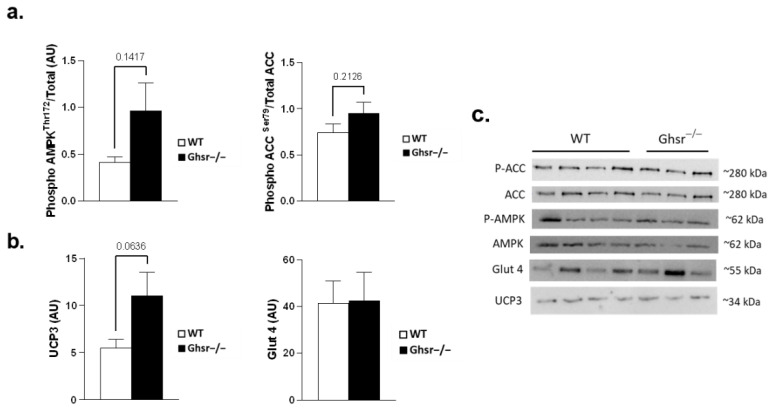
Effects of GHS-R ablation on protein markers of metabolic functions in aging muscle. Gastrocnemius muscle of old (18–26 m) WT (open bar, *n* = 4) and *Ghsr^−/−^* (black bar, *n* = 3) mice is shown. (**a**) Activation of AMPK and ACC expressed as phosphorylated to total protein ratios. (**b**) Total Protein expression of UCP3 and Glut 4 in gastrocnemius muscle. (**c**) Representative images for AMPK, ACC, UCP3 and Glut 4. Two-tailed Student’s *t*-tests were performed, and data are presented as means ± standard error.

**Figure 4 genes-13-01368-f004:**
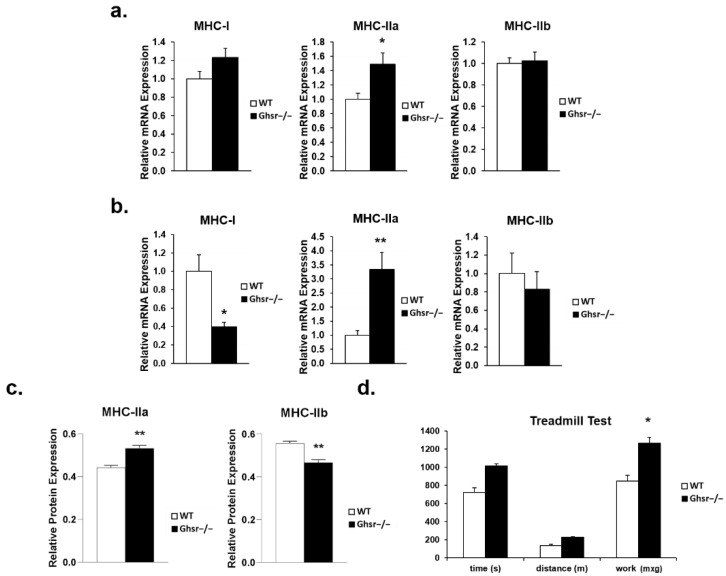
Expression of myosin heavy chain subtypes and treadmill work output of old *Ghsr^−/−^* mice. Old (16–24 m) WT (open bar) and *Ghsr^−/−^* (closed bar) mice were used in this set of experiments. (**a**) Relative mRNA expression of myosin heavy chain in gastrocnemius (*n* = 9). (**b**) Relative mRNA expression of myosin heavy chain in soleus muscle (*n* = 6). (**c**) Protein expression of myosin heavy chain isoforms of MHC-IIa and MHC-IIb relative to total Myosin heavy chain expression in old WT (*n* = 4) and *Ghsr^−/−^* (*n* = 3) mice. (**d**) Treadmill test—time, distance and work of old WT and *Ghsr^−/−^* mice (*n* = 10). S: seconds of time; m: meters of distance; mxg: work performed during treadmill. 18s and β-actin house-keeping genes were used as internal controls in qPCR analysis. Two-tailed Student’s *t*-test were performed, and data are presented as means ± standard error. * *p* < 0.05, ** *p* < 0.001, *Ghsr^−/−^.* vs. WT.

**Figure 5 genes-13-01368-f005:**
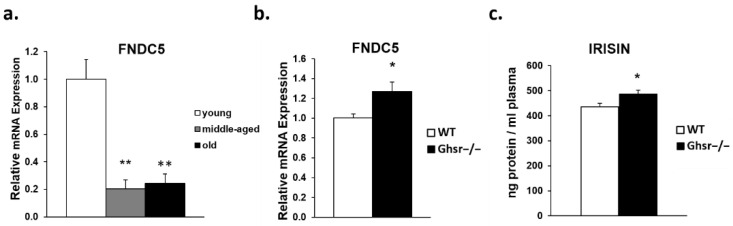
*FNDC5* expression in gastrocnemius muscle and plasma irisin in the circulation. (**a**) *FNDC5* mRNA expression in young (4–5 m), middle-aged (12–14 m) and old (18–26 m) WT mice (*n* = 6). (**b**) *FNDC5* mRNA expression in gastrocnemius muscle of old WT (open bar) and *Ghsr^−/−^* (filled bar) mice. (**c**) Plasma irisin levels in old WT (open bar) and *Ghsr^−/−^* (filled bar) mice. 18s and β-actin house-keeping genes were used as internal controls in qPCR analysis. One-way ANOVA was performed in (a) and Student’s *t*-tests were performed for (b) and (d). Data are presented as means ± standard error. * *p* < 0.05, ** *p* < 0.001, middle-aged or old mice vs. young mice, or *Ghsr^−/−^.* vs. WT.

## Data Availability

Not Applicable.

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
