# Peer review of "Ablation of Ghrelin Receptor Mitigates the Metabolic Decline of Aging Skeletal Muscle"

_genes, 2022, doi:10.3390/genes13081368_

Round 1
Reviewer 1 Report
I was honored to review a manuscript entitled Ablation of Ghrelin Receptor Mitigates the Metabolic Decline of Aging Skeletal Muscle. In this manuscript, the authors have found that the ablation of ghrelin receptors may have beneficial effects on the metabolic deterioration of skeletal muscle induced by aging. The results were novel and interesting. However, there are a few key points that need to be revised. Some comments and suggestions are as follows.
Abstract
- Line 22: It’d be better not to mention adipose tissue as it was not covered in this paper.
- Line 29: The conclusion seems somewhat exaggerated. The effect of irisin is still controversial, and the authors have not experimentally confirmed a relationship between irisin and the metabolic changes induced by GHS-R ablation.
Introduction
- Please explain the role of irisin in the metabolic regulation of skeletal muscle itself.
Materials and Methods
- Please add the number of animals used in each group.
- During the experimental period, did the authors measure the intake? This result may be necessary to explain the mechanism of metabolic improvement in Ghsr-/- mice.
- Please specify the internal control genes used in each figure legend because the authors have indicated that they used both 18S RNA and b-actin.
- It seems that the authors used gastrocnemius and soleus muscles. However, in the method section, the soleus muscle has not been mentioned.
- The silver staining images should be given.
- The authors have stated that this paper used the two-factor ANOVA, post hoc tests, and two-tailed Student’s t-test. Therefore, it is necessary to indicate the statistical method used in each figure legend. In addition, and when appropriate, the interaction should be mentioned in the result section.
Results
- Did the authors analyze metabolic consequences associated with UCP3, SIRT1, and PGC1a or GLUT and IRS1? Gene expression results alone are not sufficient to explain the metabolic improvement due to Ghsr gene ablation.
- In the western blot results, the numbers of animals used were 4 (WT) and 3 (Ghsr-/-). How was the number of animals established?
- In Figure 4d, what the m x g stands for?
Discussion
- The relationship between UCP3, SIRT1, and PGC1a and GLUT and IRS1 should be explained in the discussion section.
- Treadmill performance is related to aerobic exercise. However, in the results of this paper, Ghsr-/- mice attenuated aging-induced slow-type muscle increase while improving treadmill performance. Please discuss this more.
- In the introduction section, the authors mentioned that aging is related to sarcopenia, the loss of muscle mass. So, could these results be related to age-related sarcopenia?
Reviewer 2 Report
Comments
This team has published a couple articles regarding Ghrelin receptor effects on inflammation and metabolic impairment in aging tissues. Here, the authors have suggested that ghrelin receptor has some negative effects on metabolism in aging muscle. They try to show some genetic backgrounds of Ghsr ablation in aging skeletal muscle and determine whether Ghsr regulates the muscular functions via irisin. This group already published a paper regarding that knockout of the GHS-R in rodents improves insulin sensitivity during aging by regulating fat metabolism in adipose tissues. They previously reported that Ghsr null mice show lower daily food intake, total locomotor activity, and higher oxygen consumption (VO2). Some of the results are progressive, but too narrow to explain the mechanism implicated in GHS-R and irisin axis. Little are new or novel as well as many gaps are behind their insistence.
Concerns are followed.
1. In fig 1, genetics of young vs middle vs old regarding mitochondrial biogenesis and energy metabolism in gastrocnemius muscle was analyzed. Levels of proteins or activities were needed to increase data significance.
2. Please mention that there any difference in diet, the reason that the lipid content decrease, and overall weight changes.
3. In fig 2, effects of GHS-R ablation on metabolic dysfunction in ‘old’ Ghsr-/- mice were assessed. Please show them in young mice (4 month), too.
4. In fig 2, was there a quantitative difference in mitochondria according to the PGC-1alpha change in each skeletal muscle of GHS-R null mice?
5. In fig 4, authors need to show more evidence regarding muscle types in WT vs Ghsr-/- mice.
6. In fig 5b, please show the leves of FNDC5 and Irisin in young or middle Ghsr null mice,
7. In Q6 regard, what expected in acute exercise or continuous exercise.
8. It is reported that the αV integrins are the receptors for irisin. Do Ghsr-/- mice have higher activity in skeletal muscle too?
9. It would be desirable to conduct an insulin tolerance experiment of the Ghsr-/- mice.
Author Response
Great appreciate your insightful critiques, please see attachment.
